# Spatio-Temporal Transcriptional Dynamics of Maize Long Non-Coding RNAs Responsive to Drought Stress

**DOI:** 10.3390/genes10020138

**Published:** 2019-02-13

**Authors:** Junling Pang, Xia Zhang, Xuhui Ma, Jun Zhao

**Affiliations:** Biotechnology Research Institute, Chinese Academy of Agricultural Sciences, Beijing 100081, China; pangjunling@caas.cn (J.P.); zhangxia@caas.cn (X.Z.); maxuhui1995@163.com (X.M.)

**Keywords:** maize, lncRNA, drought stress, transcriptional profiling, tissue and development specificity

## Abstract

Long non-coding RNAs (lncRNAs) have emerged as important regulators in plant stress response. Here, we report a genome-wide lncRNA transcriptional analysis in response to drought stress using an expanded series of maize samples collected from three distinct tissues spanning four developmental stages. In total, 3488 high-confidence lncRNAs were identified, among which 1535 were characterized as drought responsive. By characterizing the genomic structure and expression pattern, we found that lncRNA structures were less complex than protein-coding genes, showing shorter transcripts and fewer exons. Moreover, drought-responsive lncRNAs exhibited higher tissue- and development-specificity than protein-coding genes. By exploring the temporal expression patterns of drought-responsive lncRNAs at different developmental stages, we discovered that the reproductive stage R1 was the most sensitive growth stage with more lncRNAs showing altered expression upon drought stress. Furthermore, lncRNA target prediction revealed 653 potential lncRNA-messenger RNA (mRNA) pairs, among which 124 pairs function in *cis*-acting mode and 529 in *trans*. Functional enrichment analysis showed that the targets were significantly enriched in molecular functions related to oxidoreductase activity, water binding, and electron carrier activity. Multiple promising targets of drought-responsive lncRNAs were discovered, including the V-ATPase encoding gene, *vpp4*. These findings extend our knowledge of lncRNAs as important regulators in maize drought response.

## 1. Introduction

Drought is a major abiotic stress that adversely impacts plant growth and productivity. Drought-induced loss in crop production at regional to global scales is becoming a growing threat to agricultural system and food security [1,2]. As a result, it is necessary to understand how plants respond to drought stress by changing metabolic, physiological, and developmental activities. To address these underlying altered activities, many previous studies focused on the dramatic reprogramming of gene expression and made great advances [3,4,5,6,7,8,9,10]. Recently, however, more emerging long non-coding RNAs (lncRNAs), as a new frontier for gene regulation, have been found to play vital roles in plant development and stress response and are drawing wider attention of researchers [11,12,13,14,15,16,17].

Long non-coding RNAs represent a large class of transcripts longer than 200 nucleotides and usually do not have or have low protein-coding potential [18,19]. Compared to protein-coding RNAs, lncRNAs are more specific to various tissues, developmental stages, and growing conditions making them promising candidates as regulators of important biological processes [17,20]. Similar to messenger RNA (mRNA), certain lncRNAs are induced during abiotic stress and thus play critical regulatory roles in plant stress response [21]. For instance, the expression of two lncRNAs, *COOLAIR* and *COLDAIR*, could be induced by cold and thus regulate the vernalization process by repressing the expression of FLOWERING LOCUS C (*FLC*) [22,23]. Recently, the identification and characterization of the lncRNA—*DRIR* (Drought induced lncRNA) in *Arabidopsis thaliana* highlight the critical role of lncRNAs in regulating plant tolerance to drought and salt stress [24]. Although some studies on the role of lncRNAs in plants have been performed, comprehensive surveys of lncRNA responses to drought stress remain limited.

With the extensive applications of high-throughput sequencing technology, it becomes easier to characterize and conduct functional analysis of lncRNAs in species. In maize, numerous lncRNAs have been identified using available expressed sequence tag (EST) and/or RNA sequencing (RNA-seq) data obtained from 30 different experiments containing 13 distinct tissues under the well-watered condition [25]. Also, several studies have conducted the lncRNA analysis in maize under drought stress [26,27]. These studies mostly focused on either a single tissue or a single growth stage. However, reports on lncRNAs involved in drought responsive regulation and addressing their spatio-temporal transcriptional dynamics are lacking. An extensive in-depth analysis of the stage-dependent and tissue-specific lncRNA expression in response to drought will help understand the role of lncRNAs in drought response with respect to developmental process of maize tissue.

In this study, we took advantage of RNA-seq data collected from three distinct tissues (ear, tassel, and leaf) across four developmental stages (V12, V14, V18, and R1) under both well-watered and drought conditions [28] and performed a genome-wide screening and functional characterization of lncRNAs in maize. Through our stringent bioinformatic pipeline, we expect to: (i) isolate a collection of lncRNAs expressed in distinct tissues at different developmental stages under both well-watered and drought conditions; (ii) identify the lncRNAs specifically responding to drought and characterize their dynamic expression patterns across the developmental stages; (iii) predict the targets of drought-responsive lncRNAs and provide insights into the potential roles in response to drought stress.

## 2. Materials and Methods

### 2.1. Transcriptome Data for Characterizing Long Non-Coding RNAs

The transcriptional data from the maize reference inbred line B73 were obtained from the NCBI BioProject with accession number assigned to PRJNA291919 [28]. These data were gathered from a total of 94 RNA-seq experiments involving three distinct tissues (ear, tassel, and leaf). These RNA-seq samples were collected at four different development stages (V12, V14, V18, and R1) under both well-watered and drought stress conditions, each with three or four biological replicates.

### 2.2. Bioinformatic Pipeline for Identifying Long Non-Coding RNAs

All RNA-seq reads of each experiment were aligned to the maize B73 reference genome (AGP v3) using HISAT2 with default parameters [29,30]. Then the aligned reads were assembled into transcripts and transcripts from all samples were merged using StringTie [31]. Based on the previously published method [25], several modified procedures were taken to identify high-confidence lncRNAs by excluding the protein-coding genes as well as other types of non-coding RNAs (ncRNAs) from the merged transcripts. Briefly, (i) known protein-coding genes from the filtered gene set (FGS) based on the gene annotation were removed from the dataset; (ii) only transcripts with length longer than 200 nt were retained for further analysis; (iii) transcripts that encode Open Reading Frames (ORFs) of 100 or less amino acids were retained as lncRNA candidates; (iv) transcripts with similarity to known proteins based on BLASTx against Swiss-Prot database were filtered out. The remaining transcripts were further evaluated using Coding Potential Assessing Tool (CPAT) (http://lilab.research.bcm.edu/cpat/) [32], which assesses the coding probability of transcripts; (v) the remaining transcripts were further filtered to exclude those that have significant alignment with housekeeping RNAs (*p* < 1.0E-10) including ribosomal RNAs (rRNAs), transfer RNAs (tRNAs), small nuclear RNAs (snRNAs), small nucleolar RNAs (snoRNAs) and signal recognition particles. The sequences of maize tRNA were downloaded from the Genomic tRNA Database (http://gtrnadb.ucsc.edu/), while the sequences of the other kind of house-keeping RNAs were downloaded from the NCBI nucleotide database (https://www.ncbi.nlm.nih.gov/nuccore/). (vi) Only lncRNAs with an expression of equal to or greater than 5 reads in at least one sample were retained for downstream analysis.

### 2.3. Transcript Level Analysis for Long Non-Coding RNAs and messenger RNAs

The read counts of transcripts were calculated using R packages: GenomicFeatures and GenomicAlignments [33]. The DESeq2 package was used to normalize the read counts among samples and to calculate Fragments per Kilobase Million (FPKM) value of each transcript [34] The differentially expressed transcripts between different developmental stages or distinct tissues were also detected by DESeq2 package. A transcript was considered to be differentially expressed if its expression value was equal to or greater than 5 reads in at least one sample and the *p*-value is less than 0.05.

### 2.4. Profiling of Spatiotemporal Gene Expression in Response to Drought

To evaluate the drought effect on lncRNA expression in any specific tissue, the differential gene expression at each developmental stage was compared between drought stress and well-watered condition, where the mean value of three replicates was used for gene expression. The Short Time-series Expression Miner (STEM) was then used to compare and visualize gene expression profiles across the four developmental stages [35]. The STEM clustering method was employed to conduct gene clustering, and the maximum unit change between time points was set to 1. The Bonferroni method was used for the correction of *p*-value under the cutoff of 0.05.

### 2.5. Prediction and Annotation of Long Non-Coding RNAs Targets Responsive to Drought

Long non-coding RNAs regulate protein-coding genes via either the *cis*- or *trans*-acting ways. In our analysis, we predicted the targets of differentially expressed (DE) lncRNAs by considering two potential interaction ways: (i) lncRNAs regulate gene expression in *cis*; (ii) potential lncRNA-mRNA physical interactions in *trans.* To predict the *cis* targets, the protein-coding genes located within 20 kb upstream or downstream of DE lncRNAs were selected. The relationship between the expression of lncRNAs and protein-coding transcripts was assessed using a spearman correlation. We only selected those pairs in which both the lncRNA and its protein-coding neighbor were differentially expressed under drought stress and had a significant correlation (Spearman’s ρ ≥ |0.8|). To investigate potential lncRNA-mRNA physical interactions in *trans*, the sequences of all the possible DE lncRNA-mRNA pairs were taken to evaluate their approximate binding free energy, deltaG (dG), using LncTar software [36]. The cutoff of normalized deltaG (ndG) < −0.2 was used to select candidate lncRNA-mRNA physical interactions. In addition, the expression correlation (spearman’s ρ ≥ |0.8|) was also considered for the final selection of lncRNA-mRNA pairs interacting in *trans*.

### 2.6. Gene Ontology

Gene ontology (GO) analysis of the targets of drought responsive lncRNAs was carried out using the online toolkit agriGO (http://systemsbiology.cau.edu.cn/agriGOv2) [37]. The minimum number of mapping entries was set to 5, and Fisher’s exact tests were used to examine the significance of accumulation against the background of the corresponding whole genome (AGP v3). GO terms with false discovery rate (FDR) less than 0.05 (Yekutieli multi-test) were defined as the significantly accumulated ones. The significant GO terms were plotted using the R package ggplot2 [38].

### 2.7. Validation of the Long Non-Coding RNA MSTRG.6838.1 and its Putative Target vpp4 (GRMZM2G028432) by Real-Time Quantitative Reverse Transcriptase-PCR

Real-time quantitative reverse transcriptase-PCR (qRT-PCR) was performed to validate the expression changes of lncRNA MSTRG6838.1 and its putative target vpp4 (GRMZM2G028432) in the maize inbred B73. Two-week-old seedlings were untreated or treated with 25% PEG6000-induced osmotic stress for 6 h and 24 h. Total RNAs were isolated from leaf samples using RNAprep Pure Plant Kit (TIANGEN BIOTECH, Beijing, China) and used for first-strand complementary DNA (cDNA) reverse transcription by a FastKing gDNA Dispelling RT SuperMix kit (TIANGEN BIOTECH, Beijing, China). PCR was performed in a 20 μL reaction volume containing primers, RealStar Green Fast Mixture (GenStar, Beijing, China) and diluted cDNA templates on the Applied Biosystems 7500 Real-Time PCR System. Gene-specific primer pairs were listed in Appendix A. The cycle threshold (CT) values, corresponding to the PCR cycle number at which fluorescence emission reaches a threshold above baseline emission, were determined and the relative fold differences were calculated by 2^−ΔΔCT^ method using the maize actin1 (GRMZM2G126010) as an endogenous reference and the untreated sample as a calibrator.

## 3. Results

### 3.1. Identification and Characterization of Maize Long Non-Coding RNAs

Total 94 RNA-seq datasets obtained from public resources represent the comprehensive transcriptome data across four developmental stages (V12, V14, V18, and R1) from three major maize tissues (ear, tassel, and leaf) that were subjected to drought stress and well-watered conditions [28]. Approximately 1.4 billion raw 50-nt single-end sequence reads were obtained from the Illumina HiSeq 2500 system. We first performed reads mapping and transcript assembling for each sample as described in Methods, then the transcripts from all the 94 samples were merged. This resulted in a total number of 108,606 assembled transcripts for further identification of lncRNAs.

Several core filtering criteria were designed to identify lncRNAs from the assembled transcripts, including setting the length of transcripts longer than 200 nt, removing transcripts having high sequence similarity to known proteins or with evidence for protein-coding potential, and filtering the ‘housekeeping’ ncRNAs [19,25,39]. Moreover, to reduce noise without losing low-abundance transcripts, we retained lncRNAs with an overall expression of at least five reads in at least one sample. Through this comprehensive filtering pipeline, we finally identified a final set of 3488 high-confidence lncRNAs expressed across the main stages of developmental transition (V12, V14, V18, and R1) in ear, leaf and tassel of maize grown under drought stress and well-watered conditions (Figure 1A; Appendix A). The total number of expressed lncRNAs in each of the three tissues across the four developmental stages was comparable. Looking into the developmental stages for all the three tissues, the total number of expressed lncRNAs in the late developmental stages (V18 and R1) tended to be higher than that of early developmental stages examined (V12 and V14), especially under drought stress (Table 1).

We characterized the features of transcripts and genes of the identified lncRNAs and compared to that of maize protein-coding mRNAs. We found that the putative length of lncRNAs were generally shorter than that of protein-coding transcripts (Figure 1B). In the class of transcripts shorter than 1 kb, the proportion of lncRNAs was significantly higher than that of protein-coding transcripts (87.8% vs. 29.5%, Chi-squared test, *p* < 2.2E-16), while in the class of transcript length longer than 2 kb, the proportion of lncRNAs was much less than that of protein-coding transcripts (1.8% vs. 23.5%, Chi-squared test, *p* < 2.2E-16). In addition, the majority of lncRNAs had fewer exons than protein-coding genes. For instance, 2859 out of 3488 (82.0%) lncRNAs contained no more than two exons, while the median exon number of protein-coding genes was four (Figure 1C). We also characterized the lncRNAs based on their locations relative to the protein-coding genes (Figure 1D). We found that 2443 lncRNAs (70%) are located more than 1 kb away from the protein-coding genes (here defined as intergenic regions) and around 18% lncRNAs are located within 1 kb of the transcription start site or termination site of the protein-coding genes (here defined as promoter-TSS or TTS). The remaining small portion of lncRNAs are either intronic (9%) or overlap with the coding sequences of protein-coding genes (3%). The distribution pattern was in agreement with previous findings that the majority of maize lncRNAs are located in the intergenic regions [25,26].

### 3.2. Long Non-Coding RNAs Responsive to Drought Stress

To understand the possible functions of lncRNAs in drought response, we performed differential expression analysis for each drought-stressed sample versus the corresponding control (e.g., leaf at V12 under drought stress versus leaf at V12 under well-watered condition). As a result, we identified 1535 differentially expressed (DE) lncRNAs upon drought treatment. The abundance and distribution of DE lncRNAs differed evidently between tissues and developmental stages. Leaf and ear tissues had relatively large number of DE lncRNAs, corresponding to 702 and 843 in total at four developmental stages analyzed, respectively, while tassel tissue had the smallest number of DE lncRNAs (Figure 2A). Compared to the three vegetative growth stages (V12, V14, and V18), reproductive stage R1 showed the highest number of DE lncRNAs in the leaf and ear tissues, two major tissues most sensitive to drought stress. These findings highlight the differential drought response of lncRNAs with respect to specific tissue and development process.

We further compared the DE lncRNAs between different tissues at each developmental stage (Figure 2B) or between different developmental stages in each tissue (Figure 2C). We found that most DE lncRNAs were distinctively expressed in the ear, leaf, and tassel tissue with less DE lncRNAs commonly shared among tissues at four developmental stages (Figure 2B). Similarly, the drought responsive lncRNAs exhibited a high degree of developmental stage specificity: DE lncRNAs that were commonly expressed between different developmental stages were relatively less than those distinctively expressed at a single stage (Figure 2C).

We also compared the expression status of lncRNAs with that of protein-coding genes under both well-watered and drought stress conditions. The number, ranging from 1 to 12, represent how many tissue types and developmental stages genes are expressed in, for example, “1” stands for genes expressed in one single tissue at a single developmental stage, while “12” refers to genes expressed ubiquitously in all three tissues at all four developmental stages examined. We used these numbers to evaluate the tissue and developmental specificity of gene expression. Under the well-watered condition, 30.3% expressed protein-coding genes (≥5 reads expressed in at least one replicate) in FGS were detected uniformly in all the 12 spatio-temporal combinations of tissue types and developmental stages, whereas that proportion in lncRNAs dropped precipitously to only 17.1%. By contrast, about 48.0% lncRNAs were expressed in no more than four tissues and developmental stages, which was higher than that of protein-coding genes (31.9%) (Figure 2D). Under drought stress, we compared the expression status of differentially expressed lncRNAs and protein-coding genes. The lncRNAs showed a clearer trend of tissue/development-specificity under drought compared to the well-watered condition. Although protein-coding gene expression also exhibited tissue/stage-specificity in response to drought, it was weaker than that of lncRNAs: the percentage of protein-coding mRNAs that exhibited differentially expressed only in a single tissue at a single developmental stage was 41.0%, while the percentage reached up to 57.8% for lncRNAs (Figure 2E).

### 3.3. The Dynamic Developmental Expression of Drought Responsive Long Non-Coding RNAs

With the STEM method, we input a list of differentially expressed lncRNAs to detect the expression patterns of drought responsive lncRNAs across development stages for each tissue. In each tissue, the differential expression values of lncRNAs under drought stress (Exp_drought_-Exp_water_) of any specific developmental stage were normalized by subtracting the expression value of the initial stage (V12). In this way, the expression trends of lncRNAs responsive to drought stress through the developmental stages were clearly exhibited. The respective lncRNAs were separated into two groups, up-regulated and down-regulated upon drought stress, to avoid the signal interference. We categorized the expression pattern of the up-regulated and down-regulated lncRNAs into “most sensitive at a later stage”, “most sensitive at an early stage”, and “patterns with varied directions” (Figure 3A). We found that the majority of the up-regulated and down-regulated lncRNAs were in the category of “most sensitive at a later stage”, among which the most significant expression changes occurred at the R1 stage. For example, the 127 up-regulated lncRNAs in ear falling into the category of “most sensitive at a later stage” displayed the most induced expression at R1 stage, accounting for 68.6% in all the significant patterns. These findings suggest that R1, as the transition from the vegetative stage to the reproductive stage, is a critical stage responding to drought stress by regulating lncRNA expression.

R1 is the first reproductive stage when silks emerge from the ear to receive pollen and begin the fertilization process. Any severe stress occurred at this growth stage can easily abort kernel and cause great yield loss [40,41,42]. Therefore, we further investigate into those lncRNAs that respond to drought specifically in ear and simultaneously have the most expression changes at the R1 stage. As a result, 132 lncRNAs were identified, which either showed up-regulated or down-regulated expression under drought (Appendix A). To clearly exhibit the expression changes of these lncRNAs, we took two drought responsive lncRNAs to show their expression changes during the development process in ear. Long non-coding RNA MSTRG.2834.1 responded to drought specifically in ear and showed the most expression changes at R1 stage. Comparing with negligible changes at V12, V14 and V18 stages upon drought stress, lncRNA MSTRG.2834.1 was significantly induced (*p* = 3.6E-4) at the stage R1 (Figure 3B). However, for lncRNA MSTRG.43642.1, drought stress barely had effect on its expression changes in the ear at V12, V14, and V18 stages, but it was significantly down-regulated (*p* = 2.5E-6) at the stage R1 (Figure 3C).

### 3.4. Functional Prediction of Drought-Responsive lncRNA Targets

The effects of lncRNAs on transcriptional regulation can be mediated by *cis*- and *trans*-acting modes, in which *cis*-acting lncRNAs regulate the expression of genes in close genomic proximity and *trans*-acting lncRNAs regulate the expression of distant genes [18]. In our analysis, we only investigated potential functions of two groups of lncRNAs, (i) lncRNAs that locally (or in *cis*) regulate gene expression by recruiting other transcription factors or processing gene transcription and/or splicing; (ii) *trans*-acting lncRNAs that interact with mRNA molecules by directly binding in a distant genomic position. Both functional modes require detectable expression links between the lncRNA-mRNA pairs [18]. Based on the large series of RNA-seq datasets, we evaluated their expressional links through correlation analysis, as previously described [43]. Besides, we took 20 kb as a cutoff to distinguish lncRNA-mRNA pairs acting in *cis* or *trans* based on previous expression quantitative trait loci (eQTL) analysis in maize [44].

First, we predicted the potential targets of DE-lncRNAs under drought stress in *cis*-regulatory relationships. Among 1535 DE-lncRNAs identified, 722 (~47%) lncRNAs were localized within 20 kb of neighboring DE-mRNA genes, of which 460 DE-lncRNAs had only one DE-mRNA gene and the other 262 DE lncRNAs had multiple associated adjacent DE-mRNA genes. This analysis generated a total of 1177 DE lncRNA-mRNA pairs, of which 124 (10.5%) DE lncRNA-mRNA pairs for 88 DE-lncRNAs and 104 DE-mRNAs showed strong correlation (spearman’s ρ ≥ |0.8|) (Appendix A). This proportion was significantly higher (Chi-square test, *p* < 2.2E-16) than that of all the DE lncRNA-mRNA pairs regardless of genomic distance (1.2%), suggesting that lncRNAs are more likely to be co-expressed with nearby protein-coding genes. In addition, 122 out of the 124 DE lncRNA-mRNA pairs showed positive correlation with each other, suggesting that most DE-lncRNAs were regulated in the same direction as their adjacent neighboring target genes.

To investigate potential lncRNA-mRNA physical interactions in *trans*, the approximate binding free energy, deltaG (dG) was calculated for each DE lncRNA-mRNA pair using LncTar software [36]. By choosing a high cutoff of normalized dG (ndG < −0.2) and applying expression correlation (spearman’s ρ ≥ |0.8|), we identified 539 candidate DE lncRNA-mRNA physical interactions. Among them, 529 lncRNA-mRNA pairs span beyond 20 kb in genomic distance (Appendix A), which supports the idea that distant lncRNAs mainly regulate gene expressions through physical interactions in *trans.* We also determined the direction of correlation coefficients for the 529 *trans* physical interactions, and found that 125 pairs (23.6%) have negative correlations. This proportion was much higher (Chi-square test, *p* = 5.1E-8) than that of *cis* regulations (2 out of 124, 1.6%).

We then conducted GO analysis on the putative targets of the DE lncRNAs. The significant GO terms of the targets of the *cis*-acting and *trans*-acting lncRNAs differed a lot. The targets of the *cis*-acting lncRNAs were substantially enriched in the biological processes of electron transport chain and oxidative phosphorylation, whereas the targets of the *trans*-acting lncRNAs were widely associated with DNA topological change, protein ubiquitination, and response to stimulus and homoiothermy. For the GO molecular function terms, both the two kinds of targets were significantly enriched in the oxidoreductase activity, indicating critical roles of these genes in response to drought stress. Moreover, targets of the *trans*-acting lncRNAs were also enriched in the molecular functions related to water binding, electron carrier activity etc. (Figure 4A). Among the targets of the lncRNAs, we also found several classical genes that may participate in the process of response to drought stress (Appendix A). For example, the gene *vpp4* (GRMZM2G028432), encoding a vacuolar (H^+^)-pumping ATPase subunit, was a putative target of an adjacent lncRNA MSTRG.6838.1. The expressions of *vpp4* and MSTRG.6838.1 were significantly correlated (spearman’s ρ = 0.87; *p* < 2.2E-16) in various tissues and development stages (Figure 4B) and they were both down-regulated under drought stress (Figure 4C). We validated their transcript abundance by qRT-PCR in leaf of maize seedlings that were untreated and treated with PEG-induced osmotic stress. Both lncRNA MSTRG,6838.1 and *vpp4* showed a decreased expression upon osmotic stress relative to their expression under control (Figure 4D). This result is largely consistent with the RNA-seq mapping data (Figure 4C), indicating that lncRNA MSTRG,6838.1 and *vpp4* could be a promising *cis*-acting pair that holds potential for co-regulation.

## 4. Discussion

Increasing evidence has shown that plant lncRNAs play critical roles in developmental process as well as stress response [21,45,46,47]. In order to reveal the expression patterns of maize lncRNAs in different tissues at distinct developmental stages in the process of response to drought stress, we performed the genome-wide analysis of lncRNAs using the published RNA-seq data from three distinct tissues (ear, tassel, and leaf) at four developmental stages (V12, V14, V18, and R1) of maize, which were grown under well-watered and drought conditions [28].

Through stringent bioinformatic pipelines, we discovered a set of 3488 high-confidence lncRNA transcripts. Compared to protein-coding transcripts, the structures of lncRNAs were less complex with shorter length and fewer exons, which were consistent with previous reports [14,25,48]. By comparative analysis of their expression levels between drought stress and well-watered condition, a total of 1535 lncRNAs were identified as drought responsive. They were specifically expressed in distinct tissues at different developmental stages. We found that very few drought responsive lncRNAs were shared among tissues at the same time point or among different developmental stages for a single tissue, indicating that drought responsive lncRNAs are highly tissue/development-specific. This spatiotemporal—specific lncRNAs have been discovered in mammals [49,50]. In maize, Li et al. used RNA-seq data obtained from 30 different experiments of 13 distinct tissues to identify lncRNAs expressed under well-watered conditions and they found that the expression of lncRNAs exhibited tissue specificity [25]. In agreement with previous studies, our results demonstrate that lncRNAs express in a tissue- and development–specific manner and this exquisite spatiotemporal expression patterns existed in drought responsive lncRNAs. Importantly, identification of lncRNAs with spatiotemporal expression will be essential for characterizing and understanding the functions and regulatory mechanisms of lncRNAs during plant development and stress response.

By capturing the expression patterns of drought responsive lncRNAs along the developmental processes, we found that most of the drought responsive lncRNAs showed the largest expression changes at late developmental stages, especially at the R1 stage. R1 represents a stage at which silks start to emerge and it is a critical transition from the vegetative to reproductive growth in maize. The observation of the majority of drought responsive lncRNAs having dramatic expression changes at R1 stage indicates that R1 might be a critical stage for lncRNAs-mediated gene regulation in drought stress response. Current knowledge agree that the reproductive stage from silking is sensitive to drought stress [51,52]. Our results provide insights into the possible functions of lncRNAs in response to drought at the R1 stage.

Long non-coding RNAs fulfill their regulatory functions mainly through the genes they regulate or interact with. Long non-coding RNAs can act either in *cis* or in *trans* to regulate protein-coding gene expression in plants [15]. By target prediction for the drought responsive lncRNAs, we identified 124 *cis* lncRNA-mRNA pairs and 539 *trans* pairs. We found that the majority of *cis* lncRNA-mRNA pairs (122 out of 124 *cis* pairs) have positive expressional correlations, while 23.6% *trans* lncRNA-mRNA pairs show negative correlations. These suggested that different regulatory mechanisms were used for the two kinds of lncRNAs. The functions of lncRNAs are largely determined by their locations [53]. The *cis*-acting lncRNAs that are close to the site of transcription can regulate nearby genes by recruiting transcription factors, chromatin organizers, or chromatin modifiers. The lncRNAs acting in *trans* could also function in several ways, including recruiting transcriptional factors or chromatin modifiers and base pairing with their targeted mRNAs to represses their translation [53,54].

Gene ontology analysis revealed that the target genes of the drought-responsive lncRNAs were largely enriched in biological processes or molecular functions associated with stress response, such as response to stimulus, oxidoreductase activity, and water binding. In addition, we discovered multiple promising genes that may function in plant drought response acting as the targets of lncRNA. One of them is *vpp4* (GRMZM2G028432), which encodes a vacuolar (H^+^)-pumping ATPase subunit. There are two types of vacuolar proton pumps in plant, vacuolar H^+^-ATPase (V-ATPase) and vacuolar H^+^-pyrophosphatase (V-PPase) [55]. Previous studies have revealed that two maize V-PPases, *ZmVPP1* and *ZmVPP5,* function in drought response [4,5]. Also, studies reported that over-expression of V-ATPase could confer tolerance to water deficit in tobacco or cotton [56,57]. These studies suggest that vacuolar proton pumps are involved in the adaptation to drought stress. Our identification of the gene *vpp4* that was differentially expressed under drought stress may indicate another candidate gene for improving dehydration tolerance in maize. Beyond this, the expression correlation between *vpp4* and its adjacent lncRNA MSTRG.6838.1 provides a new insight into lncRNA-mediated gene regulation in drought response. Further molecular genetics experiments are still needed to confirm the results and to dig out how the lncRNAs participate in the processes of response to drought stress.

## Figures and Tables

**Figure 1 genes-10-00138-f001:**
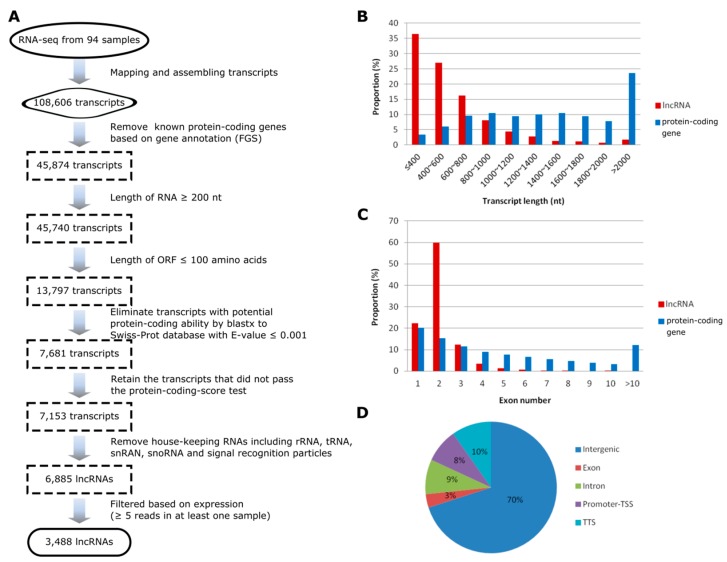
**Identification and characterization of maize long non-coding RNAs (lncRNAs).** (**A**) The pipeline for the identification of maize lncRNAs. (**B**) Comparison of the length between lncRNAs and the protein-coding transcripts. (**C**) Numbers of exons in lncRNA and protein-coding genes. (**D**) Genomic positions of lncRNAs. FGS, Filtered Gene Set; TSS, Transcription Start Site; TTS, Transcription Terminal Site.

**Figure 2 genes-10-00138-f002:**
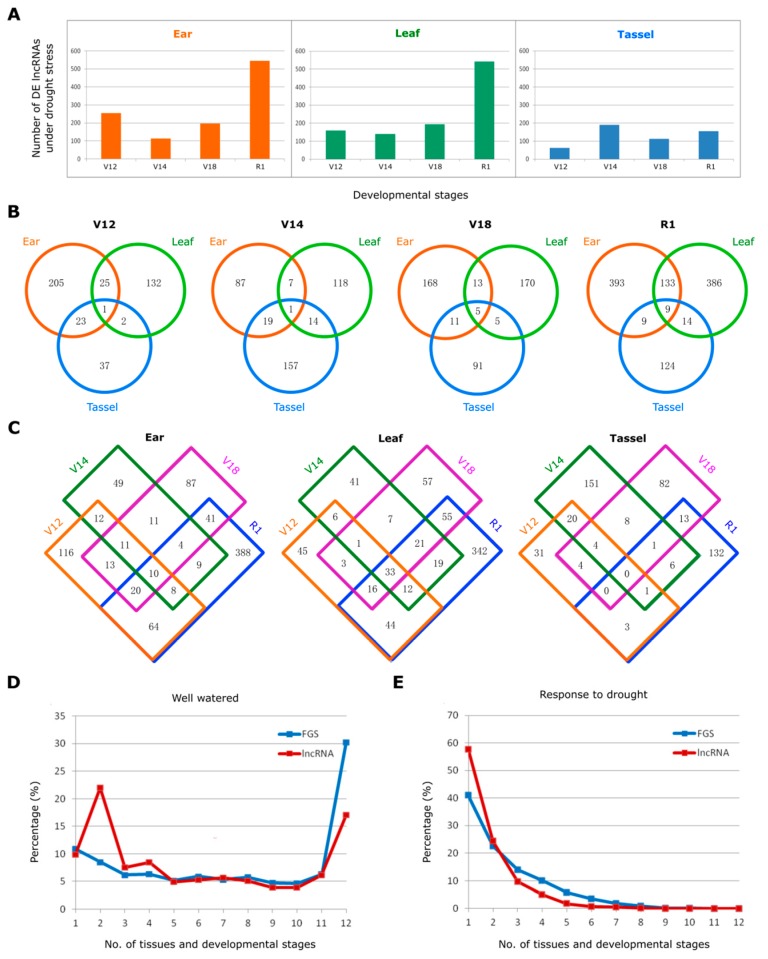
**Identification of drought responsive lncRNAs and their tissue- and developmental specificity of expression.** (**A**) Number of differentially expressed lncRNAs under drought stress in ear, leaf, and tassel. (**B**) Venn diagram showing the lncRNAs differentially expressed in the three tissues (ear, leaf, tassel) at four developmental stages (V12, V14, V18, R1). (**C**) Venn diagram showing the lncRNAs differentially expressed at the four developmental stages (V12, V14, V18, R1) in each tissue (ear, leaf, tassel). (**D**) Tissue/development-specific expression of the lncRNAs and FGS transcripts under the well-watered condition. The number on the *x*-axis, ranging from 1 to 12, represent how many tissue types and developmental stages genes are expressed in, for example, “1” stands for genes expressed in a single tissue at a single developmental stage, while “12” refers to genes expressed ubiquitously in all three tissues at all four developmental stages examined. (**E**) Tissue/development-specificity of the drought responsive lncRNAs and the drought responsive FGS transcripts. The meaning of the numbers on the *x*-axis is the same as on (**D**).

**Figure 3 genes-10-00138-f003:**
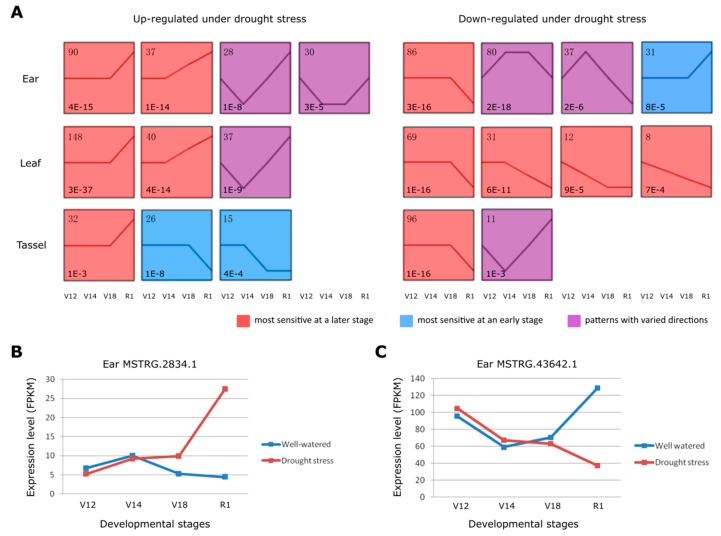
**The inductive expression patterns of drought responsive lncRNAs during development.** (**A**) The significantly enriched expression patterns of the drought responsive lncRNAs in the three tissues. The left panel shows the expression patterns of the up-regulated lncRNAs under drought stress in three tissues, while the right panel shows the down-regulated lncRNAs. The numbers in the top left corner record the number of the lncRNAs in each pattern, while the numbers in the lower left corner highlight the *p*-values of each significant pattern. Three background colors represent three large categories: red for “most sensitive at a later stage”, blue for “most sensitive at an early stage”, and purple for “patterns with varied directions”. All the expression patterns were following the timelines of maize development from V12, V14 to V18 and R1. (**B**) The expression levels (FPKM) of lncRNA MSTRG.2834.1 under both well-watered and drought stress in the four developmental stages of ear. (**C**) The expression levels (FPKM) of lncRNA MSTRG.43642.1 under both well-watered and drought stress in the four developmental stages of ear.

**Figure 4 genes-10-00138-f004:**
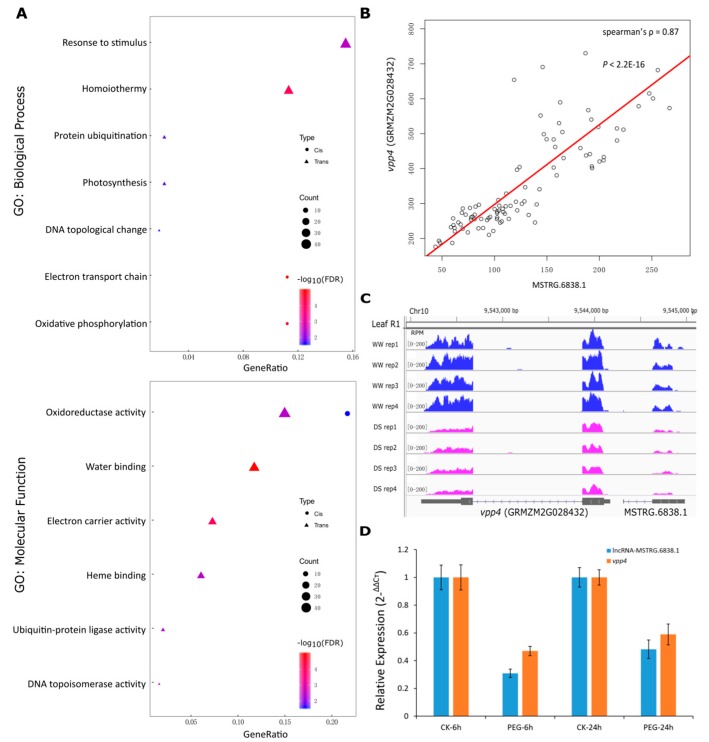
**Prediction of targets and potential functional roles of drought responsive lncRNAs.** (**A**) GO enrichments for the putative targets of the DE lncRNAs. The targets of the *cis*-acting and *trans*-acting lncRNAs were separately analyzed, and were represented using circle and triangle, respectively. The number of genes in each term was reflected by the size of the shapes, and the terms were ranked by the ratio of the lncRNA targets falling in each term to the total number of the lncRNA targets (*x*-axis). The significance of the enrichments was represented by the colors, where red represents a more significant enrichment than blue. (**B**) The expression correlation of the gene *vpp4* and its adjacent lncRNA MSTRG.6838.1. The *x*-axis marks the expression level (FPKM) of the lncRNA MSTRG.6838.1, while the y-axis marks the expression level (FPKM) of *vpp4*. (**C**) The reads coverage of the genomic regions of *vpp4* and MSTRG.6838.1 at leaf R1 stage. The first four lanes highlight the four replicates under well-watered condition, while the other four highlight the four replicates under drought stress condition. The abundance of the reads coverage was measured by RPM (Reads Per Million). (**D**) qRT-PCR validation of lncRNA MSTRG.6838.1 and its putative target *vpp4* (GRMZM2G028432) in leaf of 2-week-old seedlings of the maize inbred B73. Seedlings were untreated or treated with 25% PEG6000-induced osmotic stress for 6h and 24h. For each gene, the transcript level in the untreated plants (CK) was defined as 1. Data represent means ± standard deviation (SD) of three replicates.

**Table 1 genes-10-00138-t001:** Number of lncRNAs found in different tissues at different stages and conditions.

Stage Tissue	Well-Watered				Drought Stress				Total
	V12	V14	V18	R1	V12	V14	V18	R1	
Ear	1600	1516	1500	1753	1302	1434	1598	1814	2195
Leaf	1575	1485	1601	1107	1554	1722	1501	1828	2144
Tassel	1342	1701	2332	2026	1352	1773	2238	2218	2931
Total	2195	2323	2924	2832	2053	2388	2948	3120	3488

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
