# Peer review of "Spatio-Temporal Transcriptional Dynamics of Maize Long Non-Coding RNAs Responsive to Drought Stress"

_genes, 2019, doi:10.3390/genes10020138_

Round 1
Reviewer 1 Report
The manuscript is well written and detailed in silico analysis have been carried out which provides leads on drought inducible lncRNAs in various tissues at different growth stages in maize. However, if possible, the author could have validated the expression of some of these lncRNAs or their targets by wet lab experiment.
Other minor comments are:
Line 9: Abstract: the small letter ‘l’ in lncRNA is written as capital ‘I’, correct it >we report a genome-wide IncRNA transcriptional<
Fig 1A. Correct the spelling of ‘amino acid’ in the fourth step.
Line 147: >than 200 nt , < delete space before comma
Author Response
Response to Reviewer 1 Comments
Point 1: The manuscript is well written and detailed in silico analysis have been carried out which provides leads on drought inducible lncRNAs in various tissues at different growth stages in maize. However, if possible, the author could have validated the expression of some of these lncRNAs or their targets by wet lab experiment.
Response 1: Thanks a lot for your encouraging comments and constructive suggestion.
In the revised manuscript, we added the experimental validation of lncRNA MSTRG.6838.1 and its putative target vpp4 (GRMZM2G028432). Maize seedlings were treated with PEG-induced osmotic stress, and the relative expressions of lncRNA MSTRG.6838.1 and vpp4 in leaf were detected by qRT-PCR. The results are largely consistent with the RNA-seq data as presented in Fig.4C, showing the down-regulated transcript level of lncRNA MSTRG.6838.1 and vpp4 under PEG stress relative to their expression under unstressed condition (new Fig. 4D).
Point 2: Other minor comments are:
Line 9: Abstract: the small letter ‘l’ in lncRNA is written as capital ‘I’, correct it >we report a genome-wide IncRNA transcriptional<
Fig 1A. Correct the spelling of ‘amino acid’ in the fourth step.
Line 147: >than 200 nt , < delete space before comma
Response 2: We have corrected these typing errors, please see Line 9, Line 164, and Fig. 1A in the revised manuscript.
Reviewer 2 Report
This is a paper characterizing the population of long noncoding RNAs (lncRNAs) in maize. The authors compare lncRNA populations in response to two conditions, well watered and drought stress. We don’t actually learn very much about drought stress in this paper, except that there are differences in lncRNAs expression in response to drought stress and that one of the differently expressed lncRNAs may be targeting a vacuolar H+-pumping ATPase. Much of the paper is bean counting, describing numbers of differentially expressed lncRNAs in different organs and at different developmental stages. The section that saves the paper is the identification of the cis and trans-acting lncRNAs and their presumptive targets.
Some minor points:
Line 161: How did the authors know that the lncRNAs were full-length?
Line 166: How can the authors eliminate the possibility that some of these lncRNA with few exons are actually protein coding but show a high frequency of exon skipping?
line 169. Because the lncRNAs have fewer exons as shown in Fig. 1 C does not necessarily mean they are less complex. Although the transcripts are small, the population could be quite complex.
Fig. 3. I don’t understand how the lncRNAs that are most sensitive at an early stage can be put in the category of “up-regulated under drought stress” when it looks like they are downregulated at a later stage. The same holds for lncRNAs that are most sensitive at an early stage in the category “down-regulated under drought stress.” These lncRNAs look like they are upregulated at a later stage.
Discussion: Since authors have found negative and positive correlations between the expression of lncRNAs and their presumptive targets, the authors should briefly describe from the literature some the means by which lncRNAs RNAs are thought to either upregulate or down regulate their target genes. This is well known for short lncRNAs, but not as well known for long lncRNAs.
Author Response
Response to Reviewer 2 Comments
Point 1: This is a paper characterizing the population of long noncoding RNAs (lncRNAs) in maize. The authors compare lncRNA populations in response to two conditions, well watered and drought stress. We don’t actually learn very much about drought stress in this paper, except that there are differences in lncRNAs expression in response to drought stress and that one of the differently expressed lncRNAs may be targeting a vacuolar H+-pumping ATPase. Much of the paper is bean counting, describing numbers of differentially expressed lncRNAs in different organs and at different developmental stages. The section that saves the paper is the identification of the cis and trans-acting lncRNAs and their presumptive targets.
Response 1: Thank you for your comments.
Point 2: Line 161: How did the authors know that the lncRNAs were full-length?
Response 2: The length of the lncRNAs are putative based on the reads coverage through bioinformatic analysis. To avoid the misleading description, we changed “full length lncRNA” to “the putative length of lncRNAs”. Thank you for your correction.
Point 3: Line 166: How can the authors eliminate the possibility that some of these lncRNA with few exons are actually protein coding but show a high frequency of exon skipping?
Response 3: We admit that our current bioinformatic pipeline cannot exclude the possibility that some of these lncRNA with few exons are actually protein coding but show a high frequency of exon skipping. As other lncRNA studies have suggested, we eliminated transcripts with potential protein-coding abilities through several steps including performing blastx against Swiss-Prot database and calculating the protein-coding scores.
Point 4: Line 169. Because the lncRNAs have fewer exons as shown in Fig. 1 C does not necessarily mean they are less complex. Although the transcripts are small, the population could be quite complex.
Response 4: Thank you for this remark. We agree that “fewer exons” does not necessarily mean “less complex”. We have deleted this sentence in the revised manuscript.
Point 5: Fig. 3. I don’t understand how the lncRNAs that are most sensitive at an early stage can be put in the category of “up-regulated under drought stress” when it looks like they are downregulated at a later stage. The same holds for lncRNAs that are most sensitive at an early stage in the category “down-regulated under drought stress.” These lncRNAs look like they are upregulated at a later stage.
Response 5: In Fig. 3, we want to show the expressional trends of drought-responsive lncRNAs across the developmental stages. We divided the drought-responsive lncRNAs into two groups: “up-regulated under drought stress” and “down-regulated under drought stress”. If a lncRNA is in the group of “up-regulated under drought stress”, the value of (Expdrought - Expwater) of any specific developmental stage is positive and vice versa. To check their trends following the developmental stages, the differential expression values of lncRNAs under drought stress (Expdrought - Expwater) of any specific developmental stage were normalized by subtracting the expression value of the initial stage (V12). Because the value of (Expdrought - Expwater) of any specific developmental stage is positive when a lncRNA is in the group of “up-regulated under drought stress”, there are cases that the value of (Expdrought - Expwater) is smaller at the later stage than V12. This does not mean that it’s down-regulated at a later stage, it’s just because the increased value of the later stage is not as much as the early stage, which we called as “most sensitive at an early stage”.
Point 6: Discussion: Since authors have found negative and positive correlations between the expression of lncRNAs and their presumptive targets, the authors should briefly describe from the literature some the means by which lncRNAs RNAs are thought to either upregulate or down regulate their target genes. This is well known for short lncRNAs, but not as well known for long lncRNAs.
Response 6: Thank you for your suggestion. We added some descriptions in the Discussion and discussed possible mechanisms by which cis- and trans-acting lncRNAs regulate the target gene expression. Please see line 394-401 in the revised manuscript.
Reviewer 3 Report
Review on manuscript scientific reports genes-396888 entitled “Spatio-temporal transcriptional dynamics of maize long non-coding RNAs responsive to drought stress” Authors: Junling Pang , Xia Zhang , Jun Zhao * In the present manuscript the authors used a dataset of an maize drought stress experiment and analyzed the long non coding RNAs. The data set consisted of RNA sequencing data from maize B73 ear, tassel and leaf at four different developmental stages V12, V14, V18 and R1 from well watered control and drought stress treatment. The author used a described pipeline and identified 3488 lncRNAs. 1535 were defined as drought responsive, as they changed their transcript abundance. Further corresponding genes were included in the analysis and 653 potential lncRNA-mRNA pairs identified. Pairs were categorized according GO and vATPase vpp4 described as putative regulated target. General comment: The authors present a well written manuscript and show a well performed reanalysis of the dataset generated before. The identification of the lncRNA in maize is new and also the drought stress related changes are of importance. Nevertheless I would request the publication of the lnRNA sequences in Table S3 and S4. Further I have to say that the manuscript is very descriptive and relates to experimental data performed elsewhere. The authors raise the interesting and important point of lncRNA regulation of genes. Up to now this is very theoretical and only based on correlation. As they identified the vpp4 as promising candidate, they should be able to prove their hypothesis by testing the abundance of the lncRNA 6838.1 and associated vpp4 in different tissues or at different stresses, or stress intensities. Figure 2 D: peak at 2 should be explained. Why is a peak detected here? Does this refer to the 133 identified lncRNAs overlapping in R1 ear and leaf? Minor points: Figure 1 A: amono acides => amino acidsAuthor Response
Response to Reviewer 3 Comments
Point 1: General comment: The authors present a well written manuscript and show a well performed reanalysis of the dataset generated before. The identification of the lncRNA in maize is new and also the drought stress related changes are of importance. Nevertheless I would request the publication of the lncRNA sequences in Table S3 and S4.
Response 1: Thank you for your encouraging comments. As suggested, we have included the lncRNA sequences in Table S3 and S4.
Point 2: Further I have to say that the manuscript is very descriptive and relates to experimental data performed elsewhere. The authors raise the interesting and important point of lncRNA regulation of genes. Up to now this is very theoretical and only based on correlation. As they identified the vpp4 as promising candidate, they should be able to prove their hypothesis by testing the abundance of the lncRNA 6838.1 and associated vpp4 in different tissues or at different stresses, or stress intensities.
Response 2: We acknowledge that this manuscript is mainly descriptive due to the article nature. We aim to provide a complete dataset of maize lncRNA in drought response through our bioinformatic analysis, and hope such findings could contribute to the follow-up studies to investigate the functional roles of the identified lncRNA-mRNA pairs.
As you suggested, we took one promising candidate pair, lncRNA MSTRG.6838.1 and its putative target vpp4 (GRMZM2G028432), to validate their transcript changes in response to drought stress. Maize seedlings were treated with PEG-induced osmotic stress and the relative expressions of lncRNA MSTRG.6838.1 and vpp4 in leaf were detected by qRT-PCR. The results are largely consistent with the RNA-seq data as presented in Fig.4C, showing the down-regulated transcript level of lncRNA MSTRG.6838.1 and vpp4 under PEG stress relative to the control unstressed condition. This experimental validation has been added to the revised manuscript. But due to the limited time and difficulty in collecting diverse tissue samples, we are not able to examine their expression changes in other tissues or different stresses. We have initiated the follow-up experiments to investigate the functional roles of the identified lncRNA-mRNA pairs, including using the combined tissues and stress treatments to test their expression profiles. Thank you for this remark!
Point 3: Figure 2 D: peak at 2 should be explained. Why is a peak detected here? Does this refer to the 133 identified lncRNAs overlapping in R1 ear and leaf?
Response 3: We apologize for not making this clear. Figure 2D presents the tissue/stage specificity of lncRNAs under well-watered condition, showing a higher tissue/stage specificity of lncRNAs than protein-coding genes. Number “2” refers to lncRNAs expressed simultaneously in two tissues/stages. We looked into this particular peak at 2 and found that there are in total 260 lncRNAs, of which 224 expressed simultaneously in Tassel at R1 stage and in Tassel at V18 stage. But under drought stress, only 14 of them responded to drought simultaneously in Tassel at R1 stage and in Tassel at V18 stage (Figure 2C, right panel). This observation supports the conclusion that lncRNAs have a higher tissue/development-specificity under drought stress than under the well-watered condition.
The 133 lncRNAs you mentioned that overlapped in R1 ear and leaf (Figure 2B, right panel) are those differentially expressed under drought stress commonly in ear and leaf. We re-checked and found that they are not included in the peak at 2 of Figure 2D.
Point 4: Minor points: Figure 1 A: amono acides => amino acids
Response 4: This error has been corrected.
Round 2
Reviewer 3 Report
Review on manuscript scientific reports genes-396888 entitled
“Spatio-temporal transcriptional dynamics of maize long non-coding RNAs responsive to drought stress”
Authors: Junling Pang , Xia Zhang , Jun Zhao *
In the present manuscript the authors used a dataset of an maize drought stress experiment and analyzed the long non coding RNAs. The data set consisted of RNA sequencing data from maize B73 ear, tassel and leaf at four different developmental stages V12, V14, V18 and R1 from well watered control and drought stress treatment. The author used a described pipeline and identified 3488 lncRNAs. 1535 were defined as drought responsive, as they changed their transcript abundance. Further corresponding genes were included in the analysis and 653 potential lncRNA-mRNA pairs identified. Pairs were categorized according GO and vATPase vpp4 described as putative regulated target. The authors properly addressed the minor points from the former review and added the additional information. Unfortunately the main point raised in the former review was not properly addressed:
Although the authors performed a qPCR the result is not satisfying.
In the test and also Figure 3 the author explain the contrasting pattern in abundance of lncRNA and target RNA. In Figure 4 they show now the qRT PCR result and indicate this as relative expression.
First: by qRT-PCR only transcript abundance can be determined. Expression is tested by a different approach. Next: although the reduced abundance of vpp4 mRNA (Figure 4D) shows a responsivity on drought stress, I would expect the ncRNA reacting in the opposite direction. This detected response was investigated in seedlings. To properly verify the data retrieved by the RNA sequencing experiments the authors should test ear tissue (Figure 3) under control and stress treatment.
Author Response
Point: In the present manuscript the authors used a dataset of an maize drought stress experiment and analyzed the long non coding RNAs. The data set consisted of RNA sequencing data from maize B73 ear, tassel and leaf at four different developmental stages V12, V14, V18 and R1 from well watered control and drought stress treatment. The author used a described pipeline and identified 3488 lncRNAs. 1535 were defined as drought responsive, as they changed their transcript abundance. Further corresponding genes were included in the analysis and 653 potential lncRNA-mRNA pairs identified. Pairs were categorized according GO and vATPase vpp4 described as putative regulated target. The authors properly addressed the minor points from the former review and added the additional information. Unfortunately the main point raised in the former review was not properly addressed:
Although the authors performed a qPCR the result is not satisfying.
In the test and also Figure 3 the author explain the contrasting pattern in abundance of lncRNA and target RNA. In Figure 4 they show now the qRT PCR result and indicate this as relative expression.
First: by qRT-PCR only transcript abundance can be determined. Expression is tested by a different approach. Next: although the reduced abundance of vpp4 mRNA (Figure 4D) shows a responsivity on drought stress, I would expect the ncRNA reacting in the opposite direction. This detected response was investigated in seedlings. To properly verify the data retrieved by the RNA sequencing experiments the authors should test ear tissue (Figure 3) under control and stress treatment.
Response: Thanks again for your comments! And we feel sorry that our last reply is not satisfying.
In an effort to identify which stage is more sensitive to drought, in Figure 3, we presented the expression pattern of the drought-responsive lncRNAs (up- and down-regulated by drought) with respect to the different development stages. These results showed that drought-responsive lncRNAs have the most transition change at R1 stage, indicating R1 is a sensitive stage responding to drought stress by regulating lncRNA expression. And the two examples in Figure 3B and 3C exemplified the contrasting transcript levels of two lncRNAs expressed in R1 ear between the drought stress and well-watered condition, so no contrasting pattern in abundance of lncRNA and target RNA was shown here.
Here, lncRNA MSTRG.6838.1 and its neighboring protein-coding transcript vpp4 were identified as a cis-acting/coregulatory group, and they show the positive expression correlation (Fig. 4B), i.e. a lower expression level of lncRNA MSTRG.6838.1 in various tissues at different development stages is always associated with a lower expression of vpp4. Such positive cis-regulatory mode is generally common for cis-acting lncRNAs as documented in many studies, though lncRNA may also function to inhibit the expression of neighboring genes. The underlying mechanism is complex and may involve the transcription factors or active chromatin modifiers recruited by lncRNAs as we discussed in the manuscript. RNA-seq reads mapping in Fig. 4C showed that drought stress decreased the expression of lncRNA MSTRG.6838.1, and same expression change was observed on vpp4 which was also down-regulated by drought. This result was confirmed by qRT-PCR, lncRNA MSTRG.6838.1 has the same expression change as vpp4 in response to drought, both showing the decreased expression relative to the control, although they were examined in seedling leaf other than R1 leaf. We agree with you that testing the expression changes at R1 will be more convincing, but we don’t have tissue samples available for R1 stage right now. We will collect field-grown plants in the coming growing season and test the expressions of candidate pairs using a different approach, which will be a part of our follow-up molecular mechanism study.

Round 3
Reviewer 3 Report
In my opinion the requested experiments are required to validate the perfomed analysis.
I will not accept the statement "but we don’t have tissue samples available for R1 stage right now. We will collect field-grown plants in the coming growing season and test the expressions of candidate pairs using a different approach, which will be a part of our follow-up molecular mechanism study. "
Then the authors should wait until they have the appropropriate samples confirm their results and resubmit once this is done.